# STOCHASTIC PREDICTION OF MULTI-AGENT INTERACTIONS FROM PARTIAL OBSERVATIONS

**Chen Sun**
Google Research

**Per Karlsson**
Google Research

**Jiajun Wu**
MIT CSAIL

**Joshua B Tenenbaum**
MIT CSAIL

**Kevin Murphy**
Google Research

## ABSTRACT

We present a method that learns to integrate temporal information, from a learned dynamics model, with ambiguous visual information, from a learned vision model, in the context of interacting agents. Our method is based on a graph-structured variational recurrent neural network (Graph-VRNN), which is trained end-to-end to infer the current state of the (partially observed) world, as well as to forecast future states. We show that our method outperforms various baselines on two sports datasets, one based on real basketball trajectories, and one generated by a soccer game engine.

## 1 INTRODUCTION

Imaging watching a soccer game on television. At any given time, you can only see a subset of the players, and you may or may not be able to see the ball, yet you probably have some reasonable idea about where all the players currently are, even if they are not in the field of view. (For example, the goal keeper is probably close to the goal.) Similarly, you cannot see the future, but you may still be able to predict where the "agents" (players and ball) will be, at least approximately. Crucially, these problems are intertwined: we are able to predict future states by using a state dynamics model, but we can also use the same dynamics model to infer the current state of the world by extrapolating from the last time we saw each agent.

In this paper, we present a unified approach to state estimation and future forecasting for problems of this kind. More precisely, we assume the observed data consists of a sequence of video frames, $v_{1:T}$, obtained from a stationary or moving camera. The desired output is a (distribution over a) structured representation of the scene at each time step, $p(s_t|v_{1:t})$, as well as a forecast into the future, $p(s_{t+\Delta}|v_{1:t})$, where $s_t^k$ encodes the state (e.g., location) of the $k$'th agent and $s_t = \{s_t^k\}_{k=1}^K$.[*]

The classical approach to this problem (see, e.g., Bar-Shalom et al. (2011)) is to use state-space models, such as Kalman filters, for tracking and forecasting, combined with heuristics, such as nearest neighbor, to perform data association (i.e., inferring the mapping from observations to latent objects). Such generative approaches require a dynamical model for the states, $p(s_t|s_{t-1})$, and a likelihood model for the pixels, $p(v_t|s_t)$. These are then combined using Bayes' rule. However, it is hard to learn good generative model of pixels, and inverting such models is even harder. By contrast, our approach is discriminative, and learns an inference network to compute the posterior belief state $p(s_t|v_{1:t})$ directly. In particular, our model combines ideas from graph networks, variational autoencoders, and RNNs in a novel way, to create what we call a graph-structured variational recurrent neural network (Graph-VRNN).

We have tested our approach on two datasets: real basketball trajectories, rendered as a series of (partially observed) bird's eye views of the court; and a simple simulated soccer game, rendered using a 3d graphics engine, and viewed from a simulated moving camera. We show that our approach can infer the current state more accurately than other methods, and can also make more accurate future forecasts. We also show that our method can vary its beliefs in a qualitatively sensible way. For

---

[*]In this work, we assume the number of agents $K$ is known, for simplicity — we leave the "open world" scenario to future work.

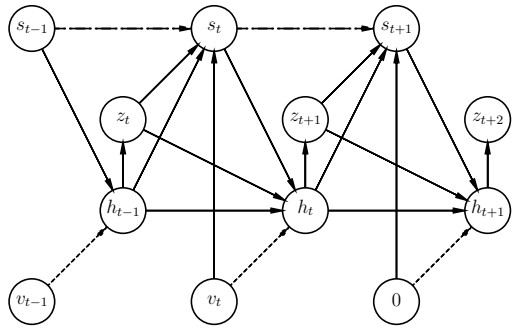

Figure 1: Illustration of visual VRNN with a single agent. Dotted edges are not used. Dashed edges are non-standard edges that we add.

example, it "knows" the location of the goalie even if it is not visible, since the goalie does not move much (in our simulation). Thus it learns to "see beyond the pixels".

In summary, our main contribution is a unified way to do state estimation and future forecasting at the level of objects and relations directly from pixels using Graph-VRNN. We believe our technique will have a variety of other applications beyond analysing sports videos, such as self-driving cars (where inferring and predicting the motion of pedestrians and vehicles is crucial), and human-robot interaction.

## 2  RELATED WORK

In this section, we briefly review some related work.

**Graph-structured and stochastic RNNs.**   There are several papers that combine standard RNNs (recurrent neural networks) with graph-structured representations, such as Chang et al. (2017); Battaglia et al. (2016); Sanchez-Gonzalez et al. (2018). There are also several papers that extend standard RNNs by adding stochastic latent variables, notably the variational RNN approach of Chung et al. (2015), as well as recent extensions, such Krishnan et al. (2017); Fraccaro et al. (2016); Karl et al. (2017); Goyal et al. (2017); Buesing et al. (2018). However, as far as we know, combining VRNNs with graphs is novel.

**Predicting future pixels from past pixels.**   There are many papers that try to predict the future at the pixel level (see e.g., (Kitani et al., 2017) for a review). Some use a single static stochastic variable, as in a conditional VAE (Kingma & Welling, 2014), which is then "decoded" into a sequence using an RNN, either using a VAE-style loss (Walker et al., 2016; Xue et al., 2016) or a GAN-style loss (Vondrick et al., 2016b; Mathieu et al., 2016). More recent work, based on VRNNs, uses temporal stochastic variables, e.g., the SV2P model of Babaeizadeh et al. (2018) and the SVGLP model of Denton & Fergus (2018). There are also various GAN-based approaches, such as the SAVP approach of Lee et al. (2018) and the MoCoGAN approach of Tulyakov et al. (2018). The recent sequential AIR (attend, infer, repeat) model of Kosiorek et al. (2018) uses a variable-sized, object-oriented latent state space rather than a fixed-dimensional unstructured latent vector. This is trained using a VAE-style loss, where $p(v_t|s_t)$ is an RNN image generator.

**Forecasting future states from past states.**   There are several papers that try to predict future states given past states. In many cases the interaction structure between the agents is assumed to be known (e.g., using fully connected graphs or using a sparse graph derived from spatial proximity), as in the social LSTM (Alahi et al., 2016) and the social GAN (Gupta et al., 2018), or methods based on inverse optimal control (Kitani et al., 2017; Lee et al., 2017). In the "neural relational inference" method of Kipf et al. (2018), they infer the interaction graph from trajectories, treating it as a static latent variable in a VAE framework. Ehrhardt et al. (2017) proposed to predict a Gaussian distribution for the uncertain future states of a sphere. It is also possible to use graph attention networks (Veličković et al., 2018) for this task (see e.g., Hoshen (2017); Battaglia et al. (2018)).

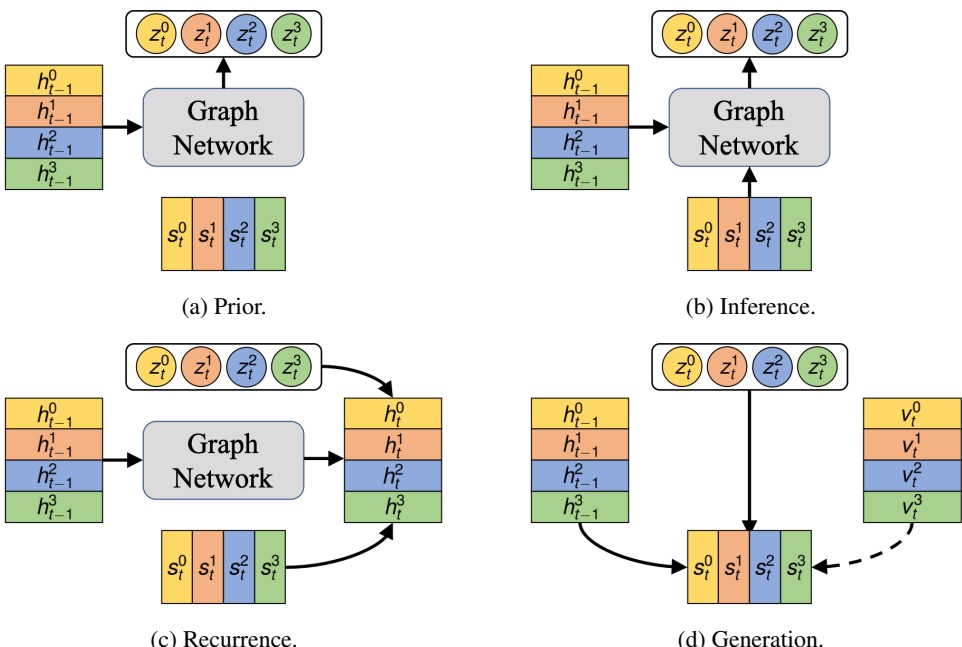

Figure 2: Illustration of the operations performed by the Graph-VRNN model, when the number of agents is 4. Dashed line is only used for the observed frames. The attention mechanism during generation stage is omitted for simplicity.

**Forecasting future states from past pixels.** Our main interest is predicting the hidden state of the world given noisy visual evidence. (Vondrick et al., 2016a) train a conditional CNN to predict future feature vectors, and several papers (e.g., (Lerer et al., 2016; Mottaghi et al., 2016; Wu et al., 2017; van Steenkiste et al., 2018; Fragkiadaki et al., 2016; Watters et al., 2017)) predict future object states. Our work differs by allowing the vision and dynamics model to share and update a common belief space; also, our model is stochastic, not deterministic. Our method is related to the "deep tracking" approach of (Dequaire et al., 2018). However, they assume the input is a partially observed binary occupancy grid , and the output is a fully observed binary occupancy grid, whereas we predict the state of individual objects using a graph-structured model. Our method is also related to the "backprop Kalman filtering" approach of (Haarnoja et al., 2016). However, they assume the structure and dynamics of the latent state is known, and they use the Kalman filter equations to integrate information from the dynamical prior and (a nonlinear function of) the observations, whereas we learn the dynamics, and also learn how to integrate the two sources of information using attention.

## 3 METHODS

### 3.1 VRNNS

We start by reviewing the VRNN approach of Chung et al. (2015). This model has three types of variable: the observed output $x_t$, the stochastic VAE state $z_t$, and the deterministic RNN hidden state $h_t$, which summarizes $x_{\leq t}$ and $z_{\leq t}$. We define the following update equations:

$$p_\theta(z_t|x_{<t}, z_{<t}) = \varphi^{\mathrm{prior}}(h_{t-1}) \qquad \text{(prior)}, \qquad (1)$$

$$q_\phi(z_t|x_{\leq t}, z_{<t}) = \varphi^{\mathrm{enc}}(x_t, h_{t-1}) \qquad \text{(inference)}, \qquad (2)$$

$$p_\theta(x_t|z_{\leq t}, x_{<t}) = \varphi^{\mathrm{dec}}(z_t, h_{t-1}) \qquad \text{(generation)}, \qquad (3)$$

$$h_t = \varphi^{\mathrm{rnn}}(x_t, z_t, h_{t-1}) \qquad \text{(recurrence)}, \qquad (4)$$

where $\varphi$ are neural networks (see section 3.4 for the details).

VRNNs are trained by maximizing the evidence lower bound (ELBO), which can be decomposed as follows:

$$\mathcal{L} = \mathbb{E}_{q_\phi(z_T|x_{\leq T}, z_{\leq T})} \left[ \sum_{t=1}^{T} \log p_\theta(x_t|z_{\leq t}, x_{<t}) - \mathrm{KL}\left(q_\phi(z_t|x_{\leq t}, z_{<t}) || p_\theta(z_t|x_{<t}, z_{<t})\right) \right]. \quad (5)$$

We use Gaussians for the prior and posterior, so we can leverage the reparameterization trick to optimize this objective using SGD, as explained in Kingma & Welling (2014). In practice, we scale the KL divergence term by $\beta$, which we anneal from 0 to 1, as in (Bowman et al., 2016). Also, we start by using the ground truth values of $x_{t-1}$ (teacher forcing), and then we gradually switch to using samples from the model; this technique is known as scheduled sampling (Bengio et al., 2015), and helps the method converge.

### 3.2 ADDING GRAPH STRUCTURE

To create a structured, stochastic temporal model, we associate one VRNN with each agent. The hidden states of the RNNs interact using a fully connected graph interaction network. (This ensures the model is permutation invariant to the ordering of the agents.) To predict the observable state $s_t^k$ for agent $k$, we decode the hidden state vector $h_t^k$ using an MLP with shared parameters. The resulting model and operations are illustrated in fig. 2.

### 3.3 CONDITIONING ON IMAGES

We can create a conditional generative model of the form $p(s_{1:t}|v_{1:t})$ by using a VRNN (where $x_t = s_t$ is the generated output) by making all the equations in section 3.1 depend on $v_t$ as input. This is similar to a standard sequence-to-sequence model, but augmented to the graph setting, and with additional stochastic noise injected into the latent dynamics. We choose to only condition the outputs $s_t$ on the visual input $v_t$, rather than making the hidden state, $h_t$, depend on $v_t$. The reason is that we want to be able to perform future prediction in the latent state space without having to condition on images. In addition, we want the latent noise $z_t$ to reflect variation of the dynamics, not the appearance of the world. See fig. 1 for a simplified illustration of our model (ignoring the graph network component). All parameters are shared across all agents except for the visual encoder; they have a shared feature extraction "backbone", but then use agent-specific features to identify the specific agent. (Agents have the same appearance across videos.) Thus the model learns to perform data association.

The key issue is how to define the output decoder, $p(s_t|z_t, h_{t-1}, v_t)$, in such a way that we properly combine the information from the current (partially observed) visual input, $v_t$, with our past beliefs about the object states, $h_{t-1}$, as well as any stochastic noise $z_t$ (used to capture any residual uncertainty). In a generative model, this step would be performed with Bayes' rule, combining the dynamical prior with the visual likelihood. In our discriminative setting, we can learn how to weight the information sources using attention. In particular, we define

$$\varphi^{\mathrm{dec}}(\mathbf{v}_t, \mathbf{h}_{t-1}, \mathbf{z}_t) = \alpha_V \cdot \varphi^{\mathrm{DV}}(\mathbf{v}_t) + \alpha_H \cdot \varphi^{\mathrm{DH}}(\mathbf{h}_{t-1}, \mathbf{z}_t) \quad (6)$$

where $\varphi^{\mathrm{DV}}$ is the visible decoder, $\varphi^{\mathrm{DH}}$ is the hidden decoder, and the $\alpha_i$ terms are attention weights, computed as follows:

$$\alpha_i = \mathrm{Sigmoid}\left(\varphi^{\mathrm{Si}}(\mathbf{v}_t, \mathbf{h}_{t-1}, \mathbf{z}_t)\right) \quad (7)$$

where $i \in \{V, H\}$.

This is similar to the gating mechanism in an LSTM, where we either pass through the current observations, or we pass through the prior hidden state, or some combination of the two. If the visual input is uninformative (e.g., for future frames, we set $v_{t+\Delta} = 0$), the model will rely entirely on its dynamics model. To capture uncertainty more explicitly, we also pass in $s_t^{\mathrm{heat}}$ into the $\varphi^{\mathrm{Si}}$ functions that compute attention, where $s_t^{\mathrm{heat}} = \mathrm{softmax}(\varphi^{\mathrm{dec}}(\mathbf{z}_t, \mathbf{h}_{t-1}, \mathbf{v}_t))$ is the set of "heatmaps" over object locations. (This is illustrated by the $s_{t-1} \rightarrow s_t$ edges in fig. 1.) We also replace the sample $\mathbf{z}_t$ with its sufficient statistics, $(\mu_t, \Sigma_t)$, computed using the state dependent prior, $\varphi^{\mathrm{prior}}(h_{t-1})$.

To encourage the model to learn to forecast future states, in addition to predicting the current state, we modify the above loss function to maximize a lower bound[†] on $\log p(\mathbf{s}_{1:T+\Delta}|\mathbf{v}_{1:T})$, computed as

---

[†]Strictly speaking, the expression is only a lower bound if $\beta = 1$. See e.g., (Alemi et al., 2018) for a discussion of the KL weighting term.

follows:

$$\sum_{t=1}^{T+\Delta T} \mathbb{E}\left[\lambda_t \log p(\mathbf{s}_t|\mathbf{z}_{\leq t}, \mathbf{v}_{\leq t}, \mathbf{s}_{\leq t})) - \beta \mathrm{KL}(\mathcal{N}_t^{\mathrm{enc}} \parallel \mathcal{N}_t^{\mathrm{prior}})\right] \tag{8}$$

where $\lambda_t = \max(t/T, 1)$ is a weighting factor that trades off state estimation and forecasting losses, $\mathcal{N}_t^{\mathrm{prior}} = \mathcal{N}(\mathbf{z}_t|\varphi^{\mathrm{prior}}(\mathbf{h}_{t-1}))$ is the prior, $\mathcal{N}_t^{\mathrm{enc}} = \mathcal{N}(\mathbf{z}_t|\varphi^{\mathrm{enc}}(\mathbf{h}_{t-1}, \mathbf{s}_t))$ is the variational posterior, the expectation is over $\mathbf{z}_{1:T}$, and where we set $\mathbf{v}_\tau = 0$ if $\tau > t$.

## 3.4 IMPLEMENTATION DETAILS

For the image encoder, we use convolutional neural networks with random initialization. In the case when there are multiple frames per step, we combine the image network with an S3D (Xie et al., 2018) inspired network, which adds additional temporal convolutions on spatial feature maps. For the recurrent model, we tried both vanilla RNNs and GRUs, and found that results were similar. We report results using the GRU model. For the graph network, we use relation networks (Santoro et al., 2017), i.e. a fully connected topology with equal edge weights. During training, both state estimation and future prediction losses are important, and need to be weighted properly. When the future prediction loss weight is too small, the dynamics model is not trained well (reflected by log-likelihood). We find that scaling the losses using an exponential discount factor for future predictions is a good heuristic. We normalize the total weights for state estimation losses and future predictions losses to be the same.

## 4 RESULTS

In this section, we show results on two datasets, comparing our full model to various baselines.

## 4.1 DATASETS

Since we are interested in inferring and forecasting the state of a system composed of multiple interacting objects, based on visual evidence, analysing sports videos is a natural choice. We focus on basketball and soccer, since these are popular games for which we can easily get data. The states here are the identities of the objects (players and ball), and their $(x, y)$ locations on the ground plane. Although specialized solutions to state estimation for basketball and soccer already exist (e.g., based on wearable sensors (Sanguesa, 2017), multiple calibrated cameras (Manafifard et al., 2017), or complex monocular vision pipelines (Rematas et al., 2018)), we are interested in seeing how far we can get with a pure learning based approach, since such a method will be more generally applicable.

**Basketball.** We use the basketball data from Zhan et al. (2018). Each example includes the trajectories of 11 agents (5 offensive players, 5 defensive players and 1 ball) for 50 steps. We follow the standard setup from Zhan et al. (2018) and just model the 5 offensive players, ignoring the defense. However, we also consider the ball, since it has very different dynamics than the players. Overall, there are 107,146 training and 13,845 test examples. We generate bird's-eye view images based on the trajectories (since we do not have access to the corresponding video footage of the game), where each agent is represented as a circle color-coded by its identity. To simulate partial observation, we randomly remove one agent from the rendered image every 10 steps. Some example rendered images can be found in fig. 3b.

**Soccer.** To evaluate performance in a more visually challenging environment, we consider soccer, where it is much harder to get a bird's-eye view of the field, due to its size. Instead most soccer videos are recorded from a moving camera which shows a partial profile view of the field. Since we do not have access to ground truth soccer trajectories, we decided to make our own soccer simulator, which we call *Soccer World*, using the Unity game engine. The location of each player is determined by a hand-designed "AI", which is a probabilistic decision tree based on the player's current state, the states of other players and the ball. The players can take actions such as kicking the ball and tackling. The ball is driven by the players' actions. Each game lasts 5 minutes. For each game, a player with the same id is assigned to a random position to play, except for the two goal keepers. We render the players using off-the-shelf human models from Unity. The identities of the players are color-coded on their shirts. The camera tracks the ball, and shows a partial view of the field at each frame. See fig. 3a for some visualizations. We create 700 videos for training, and 300 videos for test. We apply a random sliding window of 10 seconds to sample the training data. The test videos are uniformly

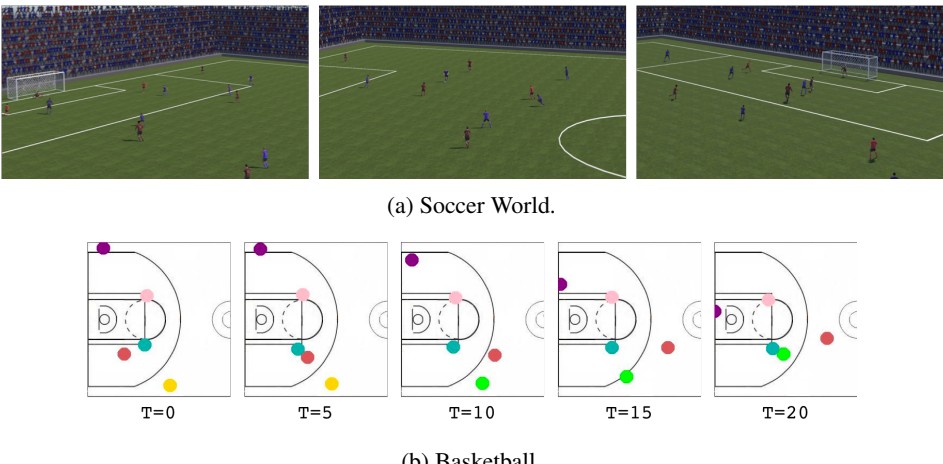

(a) Soccer World.

(b) Basketball.

Figure 3: (a) Frames from Soccer World are rendered from a moving camera. (b) Examples frames from basketball data. At $T = 0, 5$, the player (light green) who controls the ball (yellow) is not visible. At $T = 10, 15, 20$, the ball is not visible.

segmented into 10-second clips, resulting in 9000 total test examples. We plan to release the videos along with the game engine after publication of the paper. [‡]

### 4.2 EXPERIMENTAL SETUP

The first task we evaluate is inferring the current state (which we define to be the 2d location) of the all the objects (players and ball). To do this, we replace the discrete predictions with the corresponding real-valued coordinates by using weighted averaging. We then compute normalized $\ell_2$ distance between the ground truth locations and the predicted locations of all objects at each step.

The second task is predicting future states. Since there are many possible futures, using $\ell_2$ loss does not make sense, either for training or evaluation (c.f., (Mathieu et al., 2016)). Instead we evaluate the negative log-likelihood on the discretized predictions, which can better capture the multi-modal nature of the problem. (This is similar to the perplexity measure used to evaluate language models, except we apply it to each object separately and sum the results.)

For each task, we consider the following models: **Visual only**: standalone visual encoder without any recurrent neural network as backbone. For future prediction, we follow Felsen et al. (2017) and directly predict future locations from last observed visual features. **RNN**: standard RNN, the hidden states are globally shared by all agents. **VRNN**: standard VRNN. **Indep-RNN**: one RNN per agent, each RNN runs independently without any interaction. **Social-RNN**: one RNN per agent, agents interact via the pooling mechanism in Social GAN (Gupta et al., 2018). **Graph-RNN**: one RNN per agent, with a graph interaction network. **Graph-VRNN**: the full model, which adds stochasticity to Graph-RNN. All models have the same architectures for visual encoders and state decoders.

Our visual encoder is based on ResNet-18 (He et al., 2016), we use the first two blocks of ResNet to maintain spatial resolution, and then aggregate the feature map with max pooling. The encoder is pre-trained on visible players, and then fine-tuned for each baseline. We find this step important to stabilize training. For the soccer data, we down-sample the video to 4 FPS, and treat 4 frames (1 second) as one step. We consider 10 steps in total, 6 observed, 4 unobserved. We set the size of GRU hidden states to 128 for all baselines. The state decoder is a 2-layer MLP. For basketball data, we set every 5 frames as one step, and consider 10 steps as well. The size of GRU hidden states is set to 128. The parameters of the VRNN backbone and state decoders are shared across all agents, while each agent has its own visual encoder as "identifier". For all experiments, we use the standard momentum optimizer. The models are trained on 6 V100 GPUs with synchronous training with batch size of 8 per GPU, we train the model for 80K steps on soccer and 40K steps on basketball. We use a linear learning rate warmup schedule for the first 1K steps, followed by a cosine learning rate schedule.

[‡]Video samples can be found at `bit.ly/2E3qg6F`

| Method | $t=1$ | $t=2$ | $t=3$ | $t=4$ | $t=5$ | $t=1$ | $t=2$ | $t=3$ | $t=4$ | $t=5$ |
|---|---|---|---|---|---|---|---|---|---|---|
| Visual only | 0.192 | 0.190 | 0.188 | 0.188 | 0.190 | 0.081 | 0.078 | 0.077 | 0.077 | 0.080 |
| RNN | 0.189 | 0.167 | 0.164 | 0.164 | 0.163 | 0.067 | 0.061 | 0.037 | 0.047 | 0.037 |
| VRNN | 0.185 | 0.168 | 0.167 | 0.166 | 0.166 | 0.074 | 0.058 | 0.035 | 0.044 | 0.035 |
| Indep-RNN | **0.183** | 0.169 | 0.160 | 0.159 | 0.157 | 0.071 | 0.066 | 0.038 | 0.046 | 0.036 |
| Social-RNN | 0.186 | 0.167 | 0.160 | 0.156 | 0.152 | 0.069 | 0.057 | 0.036 | 0.043 | 0.037 |
| Graph-RNN | 0.189 | 0.169 | 0.159 | 0.153 | 0.147 | 0.068 | 0.055 | 0.034 | 0.041 | 0.034 |
| Graph-VRNN | 0.184 | **0.165** | **0.153** | **0.149** | **0.143** | **0.062** | **0.052** | **0.025** | **0.034** | **0.024** |

Table 1: Normalized $\ell_2$ distances to ground truth locations at different steps on (left) soccer data and (right) basketball data.

The hyper parameters, such as the base learning rate and the KL divergence weight $\beta$, are tuned on a hold-out validation set.

| Method | Ball | | Player | |
|---|---|---|---|---|
| | Visible | Hidden | Visible | Hidden |
| Visual only | 0.011 | 0.277 | 0.016 | 0.315 |
| RNN | 0.008 | 0.266 | 0.028 | 0.266 |
| VRNN | 0.009 | 0.260 | 0.029 | 0.250 |
| Indep-RNN | 0.009 | 0.279 | 0.039 | 0.262 |
| Social-RNN | 0.010 | 0.260 | 0.044 | 0.242 |
| Graph-RNN | 0.011 | 0.251 | 0.034 | 0.240 |
| Graph-VRNN | 0.006 | **0.227** | 0.013 | **0.214** |

Table 2: Average normalized $\ell_2$ distance for basketball ball and players at first 5 time steps.

| Method | Perm. Inv. | Inter- act | Stoch- astic | Soccer | Basket- ball |
|---|---|---|---|---|---|
| Prior | No | No | No | 3.0 | 1.8 |
| Visual only | No | No | No | 6.6 | 2.5 |
| RNN | No | Yes | No | 7.0 | 5.9 |
| VRNN | No | Yes | Yes | $\geq 6.7$ | $\geq 5.9$ |
| Indep-RNN | Yes | No | No | 48.9 | 14.3 |
| Social-RNN | Yes | Yes | No | 54.6 | 14.6 |
| Graph-RNN | Yes | Yes | No | 60.9 | 15.0 |
| Graph-VRNN | Yes | Yes | Yes | $\geq \mathbf{61.1}$ | $\geq \mathbf{18.4}$ |

Table 3: Future prediction performance as measured by log-likelihood ratio over random guess.

## 4.3 QUANTITATIVE RESULTS

**Basketball.** The right half of Table 1 shows the average normalized $\ell_2$ distance between the true and predicted location of all the agents for the basketball data. (Error bars are not shown, but variation across trials is quite small.) We see that the Graph-VRNN error generally goes down over time, as the system integrates evidence from multiple frames. (Recall that one of the agents becomes "invisible" every 10 frames or 2 steps, so just looking at the current frame is insufficient.)

The visual-only baseline has more or less constant error, which is expected since it does not utilize information from earlier observations. All other methods outperform the visual-only baseline. We can see that stochasticity helps (VRNN better than RNN, Graph-VRNN better than Graph-RNN), and Graph-RNN is better than vanilla RNN. As most of the players are visible in the basketball videos, we also report performance for visible and hidden (occluded) agents in Table 2. As expected, the $\ell_2$ distances for visible agents are very low, since the localization task is trivial for this data. When the agents are hidden, Graph-VRNN significantly outperforms other baselines. Again, we observe that stochasticity helps. We also find that graph network is a better interaction model than social pooling, both outperform Indep-RNN.

**Soccer.** The left half of Table 1 shows the average normalized $\ell_2$ distance between the true and predicted location of all the agents for the soccer data as a function of time. The results are qualitatively similar to the basketball case, although in this setting the $\ell_2$ distances are higher since the vision problem is much harder. However, the gains from adding stochasticity to the Graph-RNN are smaller in this setting. We believe the reason for this is that the dynamics of the agents in soccer world is much more predictable than in the basketball case, because our simulator is not very realistic. (This will become more apparent in Section 4.4.)

**Forecasting.** In this section, we assess the ability of the models to predict the future. In particular, the input is the last 6 steps, and we forecast the next 4 steps. Since the future is multimodal, we do not use $\ell_2$ error, but instead we compute the log-likelihood of the discretized ground truth locations,

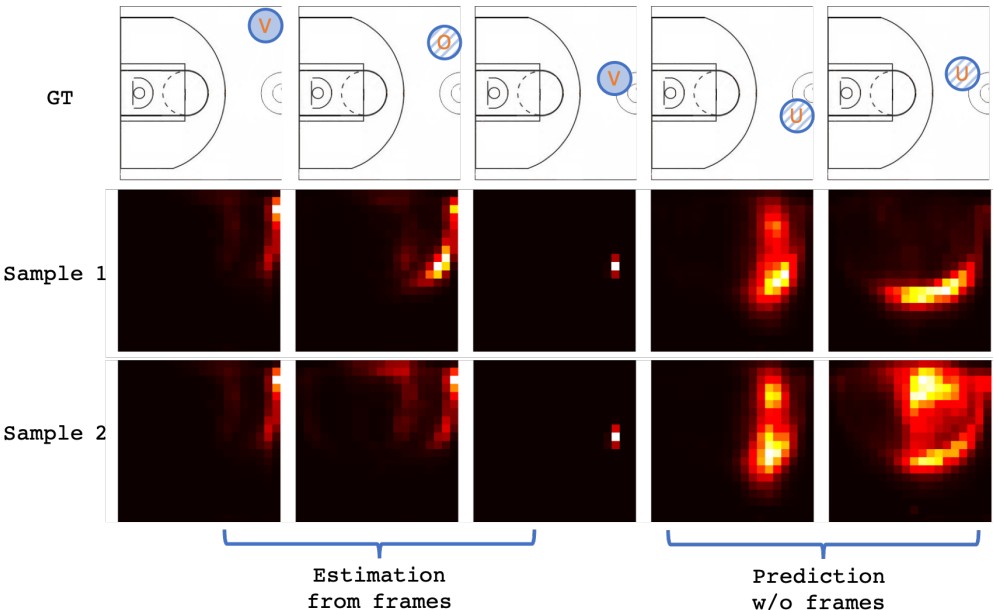

Figure 4: Belief states for one player in the basketball data across multiple different world state scenarios. We show the heatmap over possible locations based on multiple stochastic samples. The top row shows the true state of the world for reference. V is visible, O is occluded, U is unseen.

normalized by the performance of random guessing[§], which is a uniform prior over all locations. The prior baseline trains the decoder based on constant zero input vectors, thus reflecting the prior marginal distribution over locations. The visual only baseline predicts all 4 future steps based on last observed visual feature, as is in Felsen et al. (2017). The results are shown in Table 3. Not surprisingly, we see that modeling the interaction between the agents helps, and adding stochasticity to the latent dynamics also helps.

### 4.4 QUALITATIVE RESULTS

This section shows some qualitative visualizations of the belief state of the system (i.e., the marginal distributions $p(s_{t+\Delta}^k|v_{1:t})$) over time. (Note that this is different from the beliefs over the internal states, $p(z_{t+\Delta}|v_{1:t})$, which are uninterpretable.) We consider the tracking case, where $\Delta = 0$, and the forecasting case, where $\Delta > 0$. We include visualizations of sampled trajectories in the Appendix.

**Basketball.** In Figure 4, we visualize the belief state for a single agent in the basketball dataset as a function of time. More precisely, we visualize the output of the decoder, $p_t^k = \varphi^{\text{dec}}(\mathbf{v}_t, \mathbf{h}_{t-1}, \mathbf{z}_t^k)$, as a heatmap, where $\mathbf{z}_t^k \sim \varphi^{\text{prior}}(\mathbf{h}_{t-1})$ are different stochastic samples of the latent state drawn from the dynamical prior. When we roll forward in time, we sample a specific value $s_t^k \sim p_t^k$, and use this to update each $h_t^k$. When we forecast the future, we set $\mathbf{v}_t = 0$. During tracking, when visual evidence is available, we see that the entropy of the belief state reduces over time, and is consistent across different draws. However, when forecasting, we see increased entropy, reflecting uncertainty in the dynamics. Furthermore, the trajectory of the belief state differs along different possible future scenarios (values of $z_t^k$). Figure 6 shows the examples trajectories for five attacking players generated by Graph-VRNN and baseline methods. In Figure 6a, we can see that the samples generated by Graph-VRNN are consistent with ground truth for the first 5 to 6 steps (observed), and are diverse for the next 4 steps (not observed). Figure 6b shows the trajectories generated for the same example by Indep-RNN and vanilla RNN, which are cluttered or shaky.

---

[§]In the case of variational models, we report the ELBO, rather than the true log likelihood. Technically speaking, this makes the numbers incomparable across models. However, all the inference networks have the same structure, so we believe the tightness of the lower bound will be comparable.

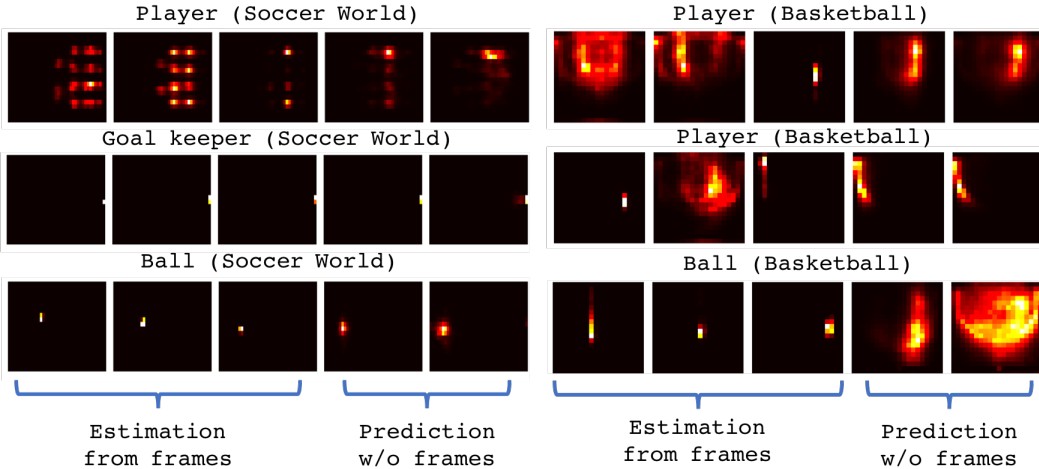

Figure 5: Heatmaps for multiple agent types for soccer data (left) and basketball data (right).

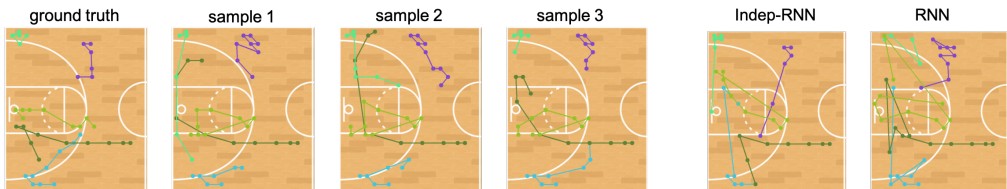

(a) Different samples generated by Graph-VRNN.        (b) Samples generated by baselines.

Figure 6: Sampled generated by Graph-VRNN and baselines for basketball data.

**Soccer.** Figure 5 (left) shows the belief states for the soccer domain for three different kinds of agents: a regular player (top row), the goal keeper (middle row), and the ball (bottom row). For the player, we see that the initial belief state reflects the 4-4-2 pattern over possible initial locations of the players. In frame 3, enough visual evidence has been accumulated to localize the player to one of two possible locations. For the future prediction, we draw a single stochastic sample (to break symmetry), and visualize the resulting belief states. We see that the model predicts the player will start moving horizontally (as depicted by the heatmap diffusing), since he is already at the top edge of the field. The goal keeper beliefs are always deterministic, since in the simulated game, the movement of the goalie is below the quantization threshold. The ball is tracked reliably (since the camera is programmed to track the ball), but future forecasts start to diffuse. Figure 5 (right) shows similar results for the basketball domain. We see that the dynamics of the players and ball are much more complex than in our soccer simulator.

## 5 CONCLUSIONS AND FUTURE WORK

We have presented a method that learns to integrate temporal information with partially observed visual evidence, based on graph-structured VRNNs, and shown that it outperforms various baselines on two simple datasets. In the future, we would like to consider more challenging datasets, such as real sports videos. We would also like to reduce the dependence on labeled data, perhaps by using some form of self-supervised learning.

**Acknowledgements.** We thank Raymond Yeh and Bo Chang for helpful discussions.

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

## A.1 SAMPLED SOCCER TRAJECTORIES

Figure A1: Sampled trajectories for Soccer World for 11 "home" players.

Figure A1 shows the sampled trajectories for Soccer World. Unlike basketball, Soccer World has a moving camera with limited field of view. We observe that the trajectories for the first few observed steps are quite shaky since only a few players have been observed. We thus show the trajectories from $t = 5$. From fig. A1, we can see that the trajectories generated by RNN are very shaky. Graph-VRNN generates much better trajectories, but some of the players are assigned incorrect identities, or incorrect locations. We conjecture that this issue can be mitigated by providing longer visual inputs to the model, such that most of the players could be observed at some point of the videos.

## A.2 SAMPLED BASKETBALL TRAJECTORIES

Figure A2 and Figure A3 provide more visualizations for different sampled trajectories by Graph-VRNN, and comparison with baseline methods. The observations are consistent as in Figure 6.

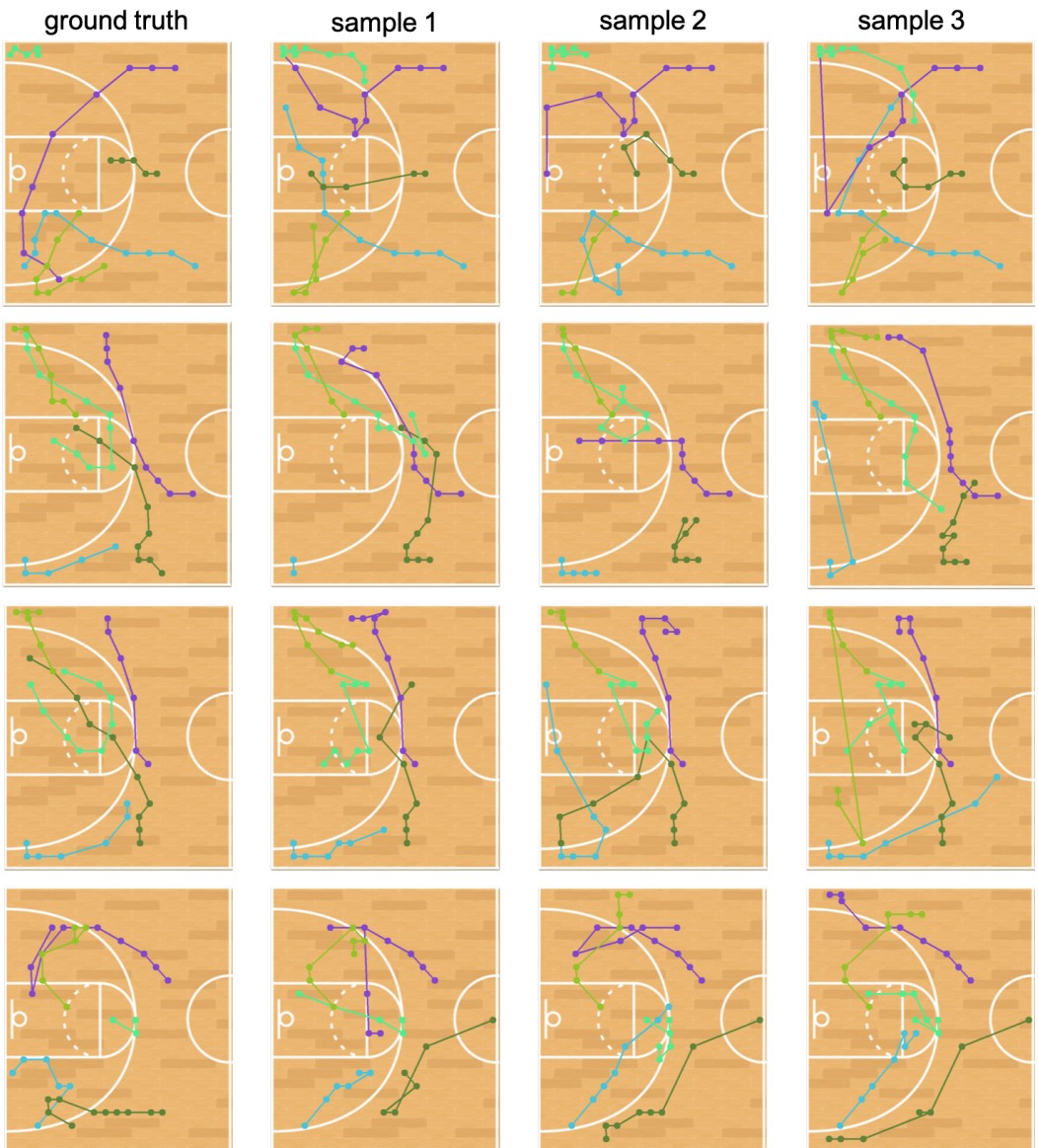

Figure A2: Sampled trajectories for 5 attacking players, generated by Graph-VRNN.

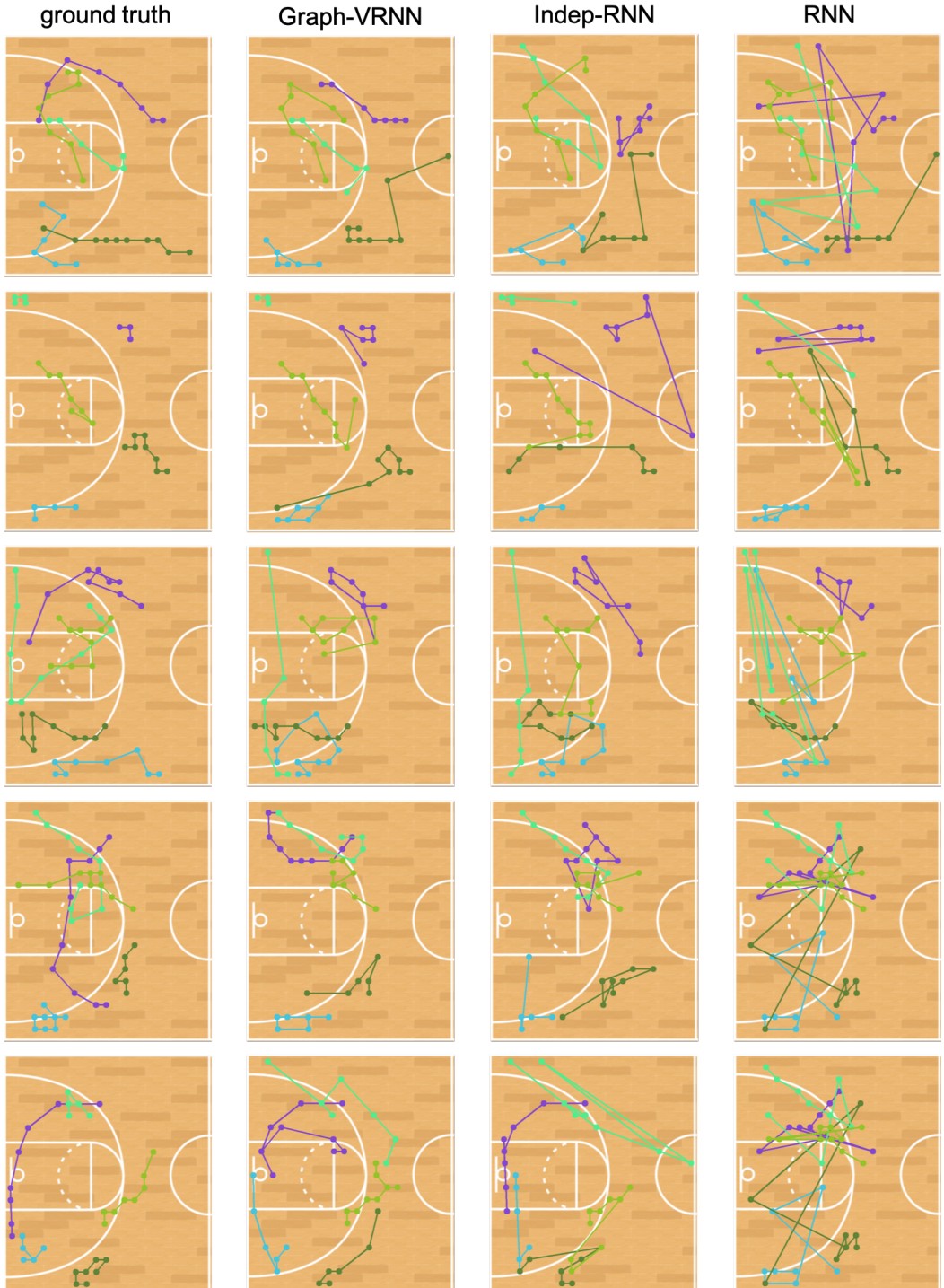

Figure A3: Sampled trajectories for 5 attacking players, generated by various baselines.

