# OpenReview forum: "Stochastic Prediction of Multi-Agent Interactions from Partial Observations"
_ICLR.cc/2019/Conference_

### Official Review · AnonReviewer2 · 2018-11-01

**Rating:** 6
**Confidence:** 4

**Review:**

1) Summary
This paper presents a graph neural network based architecture that is trained to locate and model the interactions of agents in an environment directly from pixels. They propose an architecture that is a composition of recurrent neural networks where each models a single object independently and communicate with other for the overall environment modeling. The model is trained with a variational recurrent neural network objective that allows for stochasticity in the predictions while at the same time allows to model the current and future steps simultaneously. In experiments, they show the advantage of using the proposed model for tasks of tracking as well as forecasting of agents locations.



2) Pros:
+ Novel recurrent neural network architecture to model structured dynamics of agents in an environment.
+ Outperforms baseline methods.
+ New dataset for partially observable prediction research.

3) Cons:

Forecasting task:
- The authors argue that a discretization needs to be performed because of the many possible futures given the past, and also provide an error measure based on likelihood. However, if trajectories are actually generated from these distributions, I suspect the many possible futures generated will be very shaky. Can the authors provide trajectories sampled from this? If sampling trajectories does not make sense somehow, can the authors comment on how we can sample multiple trajectories?

Lack of baselines:
- The authors mention social LSTM and social GAN in the related work, however, no comparison is provided. From a quick glance, the authors of these papers work on trajectories. However, the “social” principle in those papers is general since it’s done from the computed feature vector. Could it have not been used on top of one of the baselines? If not, could the authors provide a reason why this is not the case?


Additional comments:
As the authors mention, it would be nice to extend this paper to an unsupervised or semi-supervised task. Here are a couple of papers that may interest you:
https://arxiv.org/abs/1804.04412
https://arxiv.org/abs/1705.02193
https://arxiv.org/abs/1806.07823

4) Conclusion
Overall, the paper is well written, easy to understand, and seems to be simple enough to quickly reproduce. Additionally, the proposed dataset may be of use for the community. If the authors are able to successfully address the issues mentioned, I am willing to improve my score.

---

> ### Author Response · Authors · 2018-11-27
> **Our response**
>
> We appreciate your constructive feedback, and very useful references!
>
> (1) Forecasting task:
> We provide the sampled trajectories in Figure 5 and Appendix. In particular, Figure 5(a) and Figure A2 show the multiple samples generated by Graph-VRNN. We observe the trajectories are relatively stable. For soccer data, since the perception task is more challenging and many players are not observed, we find the belief states to be uncertain for the first several steps (having more observed steps would help in this case). For basketball data, we find that the belief states for players are usually stable, but the ball is more uncertain (bottom right row of Figure 4). We suspect that it’s due to the movement of ball is much faster (which may be addressed by using a higher FPS).
>
> (2) Lack of baselines:
> As pointed out by the reviewer, Social-LSTM and Social-GAN work with trajectory data by default. However, their social-pooling mechanism can be used as an alternative to the relation network used in Graph-RNN. We use the pooling mechanism from Social-GAN, which is more recent and has no additional hyper parameters. We find it to perform slightly worse than Graph-RNN (which uses Relation Networks), but better than Indep-RNN. Note that the graph network module is a building block in our model, and RN can be replaced by other graph network architectures.

---

> > ### Comment · AnonReviewer2 · 2018-12-03
> > **Thank you**
> >
> > Thanks for addressing my comments. The social pooling mechanism improves Indep-RNN as expected, however, as you show, it's not better than your method. This makes the results stronger. Additionally, the plotted trajectories shine light on the behavior of the trajectories. The trajectories are still better than the baselines after this additional information. Given the authors' response, I have increased my score. It will be nice to see this work take a semi-supervised or unsupervised route in the future :)

---

> > > ### Author Response · Authors · 2018-12-04
> > > **Thank you**
> > >
> > > Dear Reviewer 2,
> > >
> > > That’s great to hear. We’d like to thank you again for your very constructive comments, which have helped us improve the quality of the paper significantly.

---

### Official Review · AnonReviewer1 · 2018-11-02
**Interesting formulation; Need more evaluation**

**Rating:** 6
**Confidence:** 4

**Review:**

Summary: The paper proposes a method to predict the future state-spaces in a multi-agent system by combing the visual and temporal information using a mixed blend of Graph-Networks+VAE+RNN (G-VRNN) formulation. The proposed approach is evaluated on two sports datasets: (1). basketball sequences; (2). soccer sequences. The authors show how the overall formulation is better than each of individual components.

Pros:

1. the multi-agent setting is interesting, very natural, and has potential for many applications.

2. formulation encodes information about different aspects: agents, location, temporal activities, and each agent's relation to other.

Cons:

1. The current evaluation is contrived.

(a). the task for future state prediction in current basket-ball and soccer sequence is not very clear. A gaussian distribution defined with 'time' as standard deviation could give similar results?

(b). no comparison with the existing approaches? I think the work of Felson et al. ICCV'17 is relevant for the given paper, and so it would be ideal to do evaluation on the datasets used in their work, and if possible compare the different baselines that they have used.

(c). the goal is to predict the future state of an agent in a multi-agent setting, but it is not clear from the evaluation as how the presence of multiple agents influence the behavior of an individual.

(d). a better way to demonstrate the future state-spaces could be through trajectory of ball or players (similar to ones shown by Walker et al ECCV'16, CVPR'14). The current qualitative analysis is not sufficient to understand what is happening in the proposed pipeline.

(e). more challenging cases to demonstrate the proposed approach  -- consider any multi-person tracking dataset, and use the proposed formulation to predict multiple trajectories (and hence state-spaces at varying time) for the people. An amazing result could be shown as how a person changes trajectory as a group of people pass by.

2. The running example of 'location of goalie' is ambiguous. By design, goalie has to be near the goal post. Even if there is no visual information or any other information, one can safely say this thing?

Overall I think the work has the potential to be on something really interesting. However, I think it needs solid experiments and is not yet ready for publication.

---

> ### Author Response · Authors · 2018-11-27
> **Our response**
>
> Thank you for your helpful feedback and suggested baselines!
>
> (1) Future prediction task:
> The gaussian distribution with 'time' as standard deviation is unlikely to work well in team sports, where there are clear patterns of tactics. It is also unable to generate the trajectories as shown in Figure 5(a), Figure A1 and Figure A2.
>
> (2) Comparison with Felson et al.:
> We agree that Felson et al. ICCV'17 is relevant and added this baseline to revision. We decide to compare against the FCN method from this paper, as the other method is based on hand-crafted patterns of players’ relative locations, and not suitable for data with heavy occlusions (e.g. due to camera FOV). The FCN method is very similar to our visual encoder, except that it uses the last observed visual features to predict future states (rather than learning a prior distribution). We can see from Table 2 “visual-only” row that the approach is significantly worse than Graph-VRNN.
>
> (3) Trajectory visualizations:
> We have added qualitative evaluations of the sampled trajectories, which show that our method generates higher quality trajectories than baseline methods. Figure 5(a) and Figure A2 illustrate diverse trajectories generated by Graph-VRNN, which show there are collaborative behaviors of different players. Figure A3 compares Graph-VRNN with Indep-RNN and single RNN, which illustrates the importance of having an interaction module.
>
> (4) Goalie example:
> Yes a location prior would work very well for goalie, we will clarify this in text.

---

### Official Review · AnonReviewer3 · 2018-11-04
**Supervised learning model, experiment results are weak.**

**Rating:** 6
**Confidence:** 4

**Review:**

The authors propose Graph VRNN. The proposed method models the interaction of multiple agents by deploying a VRNN for each agent. The interaction among the agents is modeled by the graph interaction update on the hidden states of the VRNNs. The model predicts the true state (e.g., location) of the agent via supervised auto-regressive learning. The proposed model can improve this estimation from partially-observed visual observations. In the experiment, the authors apply the proposed method to Basketball and Soccer data to model the positions of the players.

The paper is clearly written. However, Section 3.2 needs to be elaborated more because using graph interaction update in VRNN is one of the main contributions.

I see two main weaknesses. The first is that the states are learned by supervised learning where obtaining the state label (i.e., the agent locations) is very expensive. Indeed, the authors had to develop their own soccer game to obtain these labels. The second weakness is the weak/inconsistent experiment results. It seems not clear whether having the graph structure or stochastic modeling is really helping or not. For example, for basketball experiment, Graph-RNN works poorly. And, for soccer, Graph-VRNN works just as good as Graph-RNN. The authors explained that this is due to the simplicity of the player behavior (not much stochastic), but the result in Table 2 shows good performance for Graph-VRNN for future prediction task. All these make it difficult to buy the claimed argument. It is also a limitation that the model requires to know and fix the number of agents.

As minor comments,

- in Table 1. Graph-RNN works better for soccer t=4, but not indicated in bold.
- Having a single RNN baseline will be helpful to compare with Graph-RNN.
- It is confusing to call s_t a belief state because it is observed not latent.
- In the qualitative results, I think it can be compared to the heatmap of true distribution.

I think the following papers needs to be discussed as related works.
- https://arxiv.org/pdf/1806.01242.pdf
- https://arxiv.org/pdf/1802.03006.pdf

---

> ### Author Response · Authors · 2018-11-27
> **Our response**
>
> Thank you for your detailed feedback! We have uploaded a revision to address your concerns:
>
> (1) Weak/inconsistent results:
> We found that joint training of visual encoder and (V)RNN/Graph-(V)RNN lead to suboptimal performance for all methods, which has been addressed by our modified visual encoder and pre-training mechanism. We can see from Table 1, 2 and 3 that graph structure consistently helps for both datasets and both tasks. We also observe that stochastic modeling is more useful for Graph-RNN than vanilla RNN.
>
> (2) Missing related work:
> We have added the references in the related work section.
>
> (3) RNN baseline:
> We have added this baseline, we find that Graph-RNN outperforms single RNN in both current state estimation and future state prediction tasks.
>
> (4) Comparison with true distribution:
> This is a great idea. We are looking into the possibility to conduct such comparison for soccer world. Unfortunately we cannot do it for basketball since the true distribution is unknown.

---

### Author Response · Authors · 2018-11-27
**Our general response**

We thank the three reviewers for your constructive feedback. The main contribution of this submission is a unified way to do state estimation and future forecasting at the level of objects and relations directly from pixels using Graph-VRNN. We focus on augmenting the experimental section based on your feedback. We hope that our revision addresses your concerns, in particular:

- We slightly modified the visual encoder to be first two blocks of a ResNet-18, followed by a spatial max-pooling. This encoder is now used by both soccer and basketball data. We pre-train the visual encoder to predict the states (locations) of only visible objects, and fine-tuned the encoder for different methods. We find that these modifications lead to higher accuracy for all methods and consistent behaviors of different methods for soccer and basketball data. Table 1, 2 and 3 shows that Graph-VRNN significantly outperform all baselines.

- We added three new baselines: (1) a vanilla RNN; (2) Social-RNN: Social-LSTM and Social-GAN do not work with visual inputs out of the box, but their social pooling mechanism can be used to replace the relation network we use in Graph-RNN. We use the pooling mechanism from Social-GAN, which is more recent and has no additional hyper parameters; (3) Felson et al. ICCV'17. The method is very similar to our visual only baseline, except that it directly predicts future states at multiple time steps from the last visual observation. We use the same visual encoder for a fair comparison.

- We added visualization of sampled trajectories in the main paper and also in Appendix. These visualizations show that Graph-VRNN is able to generate diverse and realistic trajectories.

- We replaced the cylinder players in soccer world with human models (driven by the same AI), which were not ready by the submission deadline.

---

### Meta-Review · Area_Chair1 · 2018-12-16
**Interesting for ICLR but can benefit from further evaluation**

**Confidence:** 3
**Recommendation:** Accept (Poster)

**Metareview:**

This paper proposes a unified approach for performing state estimation and future forecasting for agents interacting within a multi-agent system. The method relies on a graph-structured recurrent neural network trained on temporal and visual (pixel) information.

The paper is well-written, with a convincing motivation and a set of novel ideas.

The reviewers pointed to a few caveats in the methodology, such as quality of trajectories (AnonReviewer2) and expensive learning of states (AnonReviewer3). However, these issues do not discount much of the papers' quality. Besides, the authors have rebutted satisfactorily some of those comments.

More importantly, all three reviewers were not convinced by the experimental evaluation. AnonReviewer1 believes that the idea has a lot of potential, but is hindered by the insufficient exposition of the experiments. AnonReviewer3 similarly asks for more consistency in the experiments.

Overall, all reviewers agree on a score "marginally above the threshold". While this is not a particularly strong score, the AC weighted all opinions that, despite some caveats, indicate that the developed model and considered application fit nicely in a coherent and convincing story. The authors are strongly advised to work further on the experimental section (which they already started doing as is evident from the rebuttal) to further improve their paper.